# A Cohort Study on Cancer Incidence among Women Exposed to Environmental Asbestos in Childhood with a Focus on Female Cancers, including Breast Cancer

**DOI:** 10.3390/ijerph19042086

**Published:** 2022-02-13

**Authors:** Sofie Bünemann Dalsgaard, Else Toft Würtz, Johnni Hansen, Oluf Dimitri Røe, Øyvind Omland

**Affiliations:** 1Department of Clinical Medicine, Aalborg University, 9000 Aalborg, Denmark; 2Danish Ramazzini Centre, Department of Occupational and Environmental Medicine, Aalborg University Hospital, 9000 Aalborg, Denmark; elswur@rm.dk (E.T.W.); omland@dcm.aau.dk (Ø.O.); 3Danish Ramazzini Centre, Department of Occupational Medicine, University Research Clinic, Goedstrup Hospital, 7400 Herning, Denmark; 4Danish Ramazzini Centre, Department of Occupational Medicine, Aarhus University Hospital, 8200 Aarhus, Denmark; 5Danish Cancer Society Research Center, 2100 Copenhagen, Denmark; johnni@cancer.dk; 6Department of Oncology, Aalborg University Hospital, 9000 Aalborg, Denmark; olufdroe@yahoo.no; 7Department of Clinical and Molecular Medicine, Norwegian University of Science and Technology, 7491 Trondheim, Norway; 8Department of Oncology, Levanger Hospital, Nord-Trøndelag Hospital Trust, 7600 Levanger, Norway

**Keywords:** uterus cancer, cancer corpus uteri, cervical cancer, mesothelioma, environmental asbestos exposure, occupational asbestos exposure, asbestos cement factory, neighborhood asbestos exposure, cancer in women

## Abstract

Objectives: To examine the risk of cancer in former school children exposed to environmental asbestos in childhood with a focus on female cancers, including breast cancer. Methods: We retrieved a cohort of females (*n* = 6024) attending four schools located in the neighborhood of a large asbestos cement plant in Denmark. A reference cohort was frequency-matched 1:9 (*n* = 54,200) in sex and five-year age intervals. Using Danish registries, we linked information on historical employments, relatives’ employments, cancer, and vital status. We calculated standardized incidence rates (SIRs) for all and specific cancers, comparing these rates with the reference cohort. Hazard ratios were calculated for selected cancers adjusted for occupational and familial asbestos exposure. Results: For cancer of the corpus uteri (SIR 1.29, 95% CI 1.01–1.66) and malignant mesothelioma (SIR 7.26, 95% CI 3.26–16.15), we observed significantly increased incidences. Occupationally, asbestos exposure had a significantly increased hazard ratio for cancer in the cervix, however, a significantly lower risk of ovarian cancer. The overall cancer incidence was similar to that of the reference cohort (SIR 1.02, 95% CI 0.96–1.07). The risk of cancer of the lung was increased for those exposed to occupational asbestos, those with family members occupationally exposed to asbestos and for tobacco smokers. Conclusions: In our study, environmental asbestos exposure in childhood is associated with an increased risk of cancer of the corpus uteri and malignant mesothelioma in women.

## 1. Introduction

All forms of asbestos are documented as human carcinogens [1]. Male cancer incidence has been extensively investigated, especially following occupational asbestos exposure [2,3]. Some studies investigated the risks for women following occupational asbestos exposure [4], as well as domestic exposure from living with and handling the clothing of workers directly exposed to asbestos [5]. The majority of previous asbestos studies focused on malignant mesothelioma, an aggressive cancer strongly associated with exposure to asbestos. We previously found that females attending schools nearby to an asbestos factory in Denmark had more than a sevenfold risk of developing malignant mesothelioma by environmental exposure alone [6]. In Wittenoom, Australia, an asbestos mining town, former female residents exposed in the environment or in their own homes were found to have an excess cancer mortality compared with the female population of Western Australian [7]. Mortality ratios of mesothelioma were increased in both men and women with no occupational exposure to asbestos who had lived near an asbestos plant in Amagasaki City, Japan [8]. In a French study, mesothelioma was seen both in women with no identified asbestos exposure and in women from the same geographical areas who had an identified exposure to asbestos. This suggests that asbestos increased the risk of mesothelioma through environmental exposure [9]. In a Danish study on women with mesothelioma, the incidence of malignant mesothelioma in twenty parishes near asbestos-emitting facilities in Aalborg, Northern Denmark was significantly higher than in the general Danish female population [10]. The same study suggested that exposure to asbestos, whether environmental or domestic, is the main cause of malignant mesothelioma in women in Northern Denmark [10].

Based on these and other prior studies, we assume that people who lived close to an asbestos plant were exposed to environmental asbestos. The only asbestos cement plant in Denmark operated from 1928 to 1988 in the city of Aalborg in Northern Denmark. During this production period, a total of approximately 620,000 tons of asbestos (89% chrysotile) were manufactured.

The International Agency for Research on Cancer (IARC) has gathered sufficient evidence to establish an association between asbestos and certain types of cancer, including ovarian cancer [11]. Other female cancers have been less investigated as far as an association with asbestos is concerned. Nonetheless, increased risks have been observed for cancer of the uterus and cancer of the cervix [12,13]. The aim of the present study was to investigate whether a cohort of former female school children who lived and attended school near the asbestos cement plant have an increased risk of cancer in adulthood, with a focus on all female cancers, including breast cancer.

## 2. Materials and Methods

### 2.1. Population

The construction of the Aalborg school cohort was described in detail previously [6]. In brief, from the Aalborg City Archives, we retrieved 7th grade school records on former pupils born between 1940 and 1970 who attended a school located 100 to 750 m from the asbestos cement plant in Aalborg, Denmark. Since 2 April 1968, all residents in Denmark were assigned a unique 10-digit personal identification number (CPR number) from the Danish Civil Registration System (CRS). The CPR number was used to identify the former pupils and to sample a reference cohort that was frequency-matched 1:9 in sex and five-year birth year intervals. Construction of the cohorts and exclusion of subjects are shown in Figure 1.

### 2.2. Asbestos Exposure

The former school children not only went to school near the asbestos plant, they also most likely lived near the plant, especially considering that children in Denmark were enrolled in the public school closest to their residence until 2005.

Additional occupational asbestos exposure was assessed using the Nordic Occupational Cancer Study (NOCCA) job exposure matrix (JEM), which we evaluated, edited, and supplemented to make it compatible with Danish industry codes (DSE77). The NOCCA JEM characterizes asbestos exposure by estimates of the prevalence and level of exposure in four periods from 1945 until 1994. Previously, it was estimated that in the period before the Danish asbestos ban in 1986, approximately 150,000 persons were exposed to occupational asbestos, corresponding to approximately 10% of the working population [14]. For every cohortee, employment history was extracted from the Danish Supplementary Pension Fund Register (ATP) using the person’s CPR number. Since 1 April 1964, this register has kept data on all employments on company and industry levels for wage earners aged 16–66 years working a minimum of 9 h per week. A cohortee was classified as having been exposed to occupational asbestos if the exposure prevalence in the JEM exceeded 50% in at least one job in the period from April 1964 to 31 December 1994.

Moreover, another type of asbestos exposure for women living near asbestos-producing facilities is the so-called domestic or familial occupational exposure due to the cleaning of asbestos-contaminated working clothes, typically the spouse or son’s clothes [15,16]. We assessed the cohortee and family member’s occupational asbestos exposure in the same way. Family members, i.e., spouse, mother, father, siblings and children under the age of 18, were identified by their CPR number from the CRS, and their employment history was extracted from the ATP. A parent was classified as ever being exposed to occupational asbestos if exposure took place in the period from when the cohortee was born (the earliest April 1964) to his or her 18th birthday. A cohortee’s spouse and/or children were classified as ever being exposed to occupational asbestos if the cohortee was 18+ years old and under 18 years old, respectively. An individual from the school cohort was classified as exposed to environmental asbestos in the absence of both occupational asbestos exposure and familial occupational asbestos exposure.

### 2.3. Cancer

The IARC found sufficient evidence that asbestos exposure is associated with mesothelioma, cancer of the lung, larynx, and ovary, and positive associations were observed between asbestos exposure and cancer of the pharynx, stomach, and colorectum [11]. All cancers diagnosed in Denmark since 1943 were systematically registered by the Danish Cancer Registry (DCR), the coverage of which is nearly complete [17]. Through linkage of CPR numbers to the DCR, data on specific cancer types and dates of diagnosis were obtained for all school and reference cohort subjects. Follow-up for cancer started on either the date of starting school in the 7th grade or 2 April 1968, whichever came last. Follow-up ended on the date of death, emigration, or 31 December 2015, whichever came first. With the exception of non-melanoma skin cancers, all primary cancers diagnosed during the follow-up period were included in the analyses using the International Classification of Diseases system version ICD-7 (1943–1977), ICD-O (1978–2003) and ICD-10 (2004 and onwards) [17]. Below, “lung cancer” collectively encompasses cancer of the lung, bronchus, and trachea, and “ovarian cancer” collectively encompasses cancer of the ovary, fallopian tube and broad ligament.

### 2.4. Smoking

According to the IARC, tobacco smoking causes cancer of the lung; oral cavity; naso-, oro- and hypopharynx; nasal cavity and accessory sinuses; larynx; oesophagus; stomach; pancreas; colorectum; liver; kidney; ureter; urinary bladder; uterine cervix; ovary; and myeloid leukaemia [18]. Hence, tobacco smoking is an important potential confounder for these cancers. Information on smoking is usually not well-recorded in the Danish registries [19]. Cigarette smoking is the predominant risk factor for development of COPD [20]. Therefore, we used the diagnosis of COPD as a proxy for smoking history. The diagnosis of COPD has been registered nationwide in the Danish National Patient Registry (DNPR) since 1977 [21].

### 2.5. Statistical Analysis

Person years at risk were calculated for each subject according to the follow-up period and split into 5-year age and calendar time intervals. Accumulation of person years at risk ended at the date of death, emigration, or disappearance, whichever came first; or on 31 December 2015. To compare categorical variables between the school cohort and the reference cohort, we used the chi-square test. The Wilcoxon–Mann–Whitney test was used to calculate age medians. Standard incidence ratios (SIRs) with corresponding 95% confidence intervals (95% CI) were estimated as the overall number of observed versus expected cases, anticipating a Poisson distribution. For all female cancers, including breast cancer, together with malignant mesothelioma and lung cancer, we performed a regression analysis based on a Cox proportional hazards model in order to adjust for familial occupational asbestos exposure, occupational asbestos exposure, and smoking. In the regression analyses, adjustments were performed if the number of subjects in one exposure group (familial occupational asbestos exposure, occupational asbestos exposure or, for the smoking-associated cancers, the proxy for smoking (COPD diagnosis)) exceeded five subjects. Subjects without an ATP record were treated as if they were not exposed to occupational asbestos. Statistical analyses were performed using Stata 15.1 (Stata Corp LLC, College Station, TX, USA).

## 3. Results

The characteristics of the females from the Aalborg School Children Cohort are listed in Table 1. The final female school cohort consisted of 6024 former school children contributing 297,637 person years at risk with a median follow-up time of 51 years. Potential occupational asbestos exposure, either own or familial, was detected in 1633 women (27.1%) in the school cohort and in 9373 women (17.3%) in the reference cohort.

In the female school cohort, 912 former school children (15.1%) were diagnosed with at least one cancer. Of these, 238 women (26.1%) in the school cohort compared with 1574 women (19.7%) in the reference cohort were exposed to occupational asbestos or had a family member who was subject to occupational asbestos exposure (Table 2).

### 3.1. Cancer Incidence Ratios

To evaluate female cancer incidence, a standardized incidence ratio (SIR) with a 95% confidence interval was calculated for all female cancers including breast cancer, lung cancer and mesothelioma (Table 3). SIRs for all cancers are presented in Appendix A. The SIR is an estimate of the occurrence of cancer in the female school cohort, relative to what would be expected if the population had the same cancer experience as the reference cohort, designated as “normal” or average.

For the female cancers, we observed a modest but not statistically significant excess of cancer in external female genitals/vagina, cervix uteri, and other female genitals. The incidence of breast cancer was similar to that observed in the reference cohort. We observed fewer ovarian cancer cases than expected compared with the reference cohort (SIR 0.72, 95% CI 0.52–1.01). For cancer of the corpus uteri (SIR 1.29, 95% CI 1.01–1.66) and malignant mesothelioma (SIR 7.26, 95% CI 3.26–16.15) we observed a significantly increased incidence compared with the reference cohort.

In total, 1331 primary cancers were observed among the 6024 former female school children during the follow-up. The overall cancer incidence was at a level similar to that of the reference cohort (SIR 1.02, 95% CI 0.96 to 1.07). However, we observed notable differences in the incidence of specific cancer types between the school cohort and the reference cohort (Appendix A).

Among hematological cancers, a significant decrease in SIR was observed for myeloma.

An insignificant but numerically higher number of cancers was registered for lung cancer as well as for stomach, colon, and larynx cancer. In contrast, the number of cancers of the pharynx and rectum was lower than expected in the school cohort compared with the reference cohort.

### 3.2. Hazard Ratios

Table 4 presents results from the regression analysis for all female cancers, including breast cancer, together with malignant mesothelioma and lung cancer, the two cancer types with the strongest association with asbestos exposure.

The hazard ratio for cancer of the corpus uteri significantly increased in the environmental asbestos exposure cohort after adjustment for occupational and familial occupational asbestos exposure (HR 1.32, 95% CI 1.02–1.75). In the regression analysis, cohortees with occupational asbestos exposure had a significantly increased hazard ratio for developing cancer in the cervix uteri; on the other hand, they also had a significantly lower hazard ratio for ovarian cancer.

For breast cancer and cancer in the external female genitals, all hazard ratios were insignificant after adjustment for smoking and occupational and familial occupational asbestos exposure.

The Cox proportional hazard model demonstrated a statistically significant association between environmental asbestos exposure in childhood and malignant mesothelioma (HR 7.41, 95% CI (2.49–22.06). Because of few observations in the occupational and familial occupational asbestos exposure group, the hazard ratio for developing malignant mesothelioma was not adjusted.

The risk of cancer of the lung was statistically significantly, increasing both for those exposed to occupational asbestos (HR 1.38, 95% CI 1.06–1.80) and those with family members who had been subject to occupational asbestos exposure (HR 1.23, 95% CI 1.06–1.43). As expected, the risk of lung, bronchus, or trachea cancer also increased when adjusted for smoking, using COPD as a proxy (HR 3.55, 95% CI 3.05–4.15).

## 4. Discussion

By living and attending school near the asbestos plant in Aalborg, more than 6000 females were subject to environmental asbestos exposure during their childhood. After a median follow-up time of 51 years, these women had a higher incidence of mesothelioma and cancer of the corpus uteri than our reference cohort, compared to the general population.

Both the school cohort and the control population that was randomly selected from the whole of Denmark had a remarkably high proportion of potential occupational and domestic asbestos-exposed women (26.1% in the school cohort and 19.7% in the control population), indicating that there was a generally high risk for asbestos exposure to women in Denmark historically. This population is, therefore, also important to study, not only for mesothelioma, but also other cancer types that have not been sufficiently studied, including female-specific cancers.

For breast cancer, we did not find an increased incidence rate in the school cohort compared to the reference cohort. Similar results were found in the study from Wittenoom Australia, where breast cancer rates remained the same in the former asbestos factory workers and town residents as in the general Australian population [13]. In a British study, an association between occupational asbestos exposure and elevated rates of breast cancer was suggested. The British study found an increased rate of breast cancer diagnosed only in female factory workers involved in high-exposure jobs, e.g., manufacture of insulating material with a high asbestos content for more than two years [22]. In line with our results, most previous studies examining the mortality of women subjected to occupational asbestos exposure found no excess breast cancer mortality [12,23,24].

The hazard ratio for cancer of the corpus uteri was found to be significantly higher only for those who had been exposed to environmental asbestos but not for those with additional asbestos exposure from family or occupation. An increased risk of uterine cancer was previously reported for women exposed to asbestos [12,13,25]. This could be an incidental finding, since one would expect an increased risk for those also exposed to occupational asbestos. However, given the relative rarity of the disease and relatively low ratio of women exposed to occupational asbestos exposure (2.2% and 2.3% in the school and reference cohort, respectively; Table 1), the analysis is more robust in the environmental/unknown group; therefore, it could be a true finding.

The IARC monographs summarizing the evidence for the carcinogenicity of asbestos exposure concluded that there is a causal association between exposure to asbestos and ovarian cancer [11]. This conclusion was supported by studies showing that women and girls exposed to environmental asbestos had positive increases in both ovarian cancer incidence and mortality [7,26]. In our study, there is no clear explanation for the lower hazard ratio for ovarian cancer. However, similar results with a lower risk for ovarian cancer and a higher risk of cervical cancer were found for Wittenoom women compared with the Western Australian population [13]. The discrepancy could be due to population variations in competing risk factors such as genetic burden (e.g., BRCA1/2 mutations), hormone replacement therapy, and obesity, as well as protective factors such as multiple pregnancies and breastfeeding. In our study, the hazard ratio for cervical cancer for those exposed to occupational asbestos was significantly increased (HR 1.55, 95% CI 1.03–2.35). Since smoking is found to be a major cofactor for cervical human papillomavirus-driven carcinogenesis, we adjusted for smoking in the Cox regression analysis for cancer of the cervix; this did not alter the hazard ratio.

The increased risk of malignant mesothelioma in this cohort was previously reported in detail [6]. Similarly, subjects who attended grammar school in Casale Monferrato, a case very similar to Aalborg with an asbestos cement plant within the city, had an increased risk of malignant mesothelioma [27]. In a cohort from Wittenoom Australia, an increased cancer incidence in female “former Wittenoom children” was predominantly due to a significantly elevated incidence of malignant mesothelioma [28]. In a French national mesothelioma surveillance program, it was reported as likely that environmental and para-occupational asbestos exposure in women had an etiologic role in the occurrence of pleural mesothelioma because a third of all cases belonged in these exposure clusters [29]. Our results are in line with these studies and show that early exposure to environmental asbestos, even chrysotile asbestos, significantly increases the risk of cancer.

In a recent Danish study from Northern Jutland, non-occupational asbestos exposure, defined as living <10 km from an asbestos emitting industry (factory or shipyard) or domestic exposure was the main cause of malignant mesothelioma among women [10]. As to the role of familial asbestos exposure in the occurrence of malignant mesothelioma, another study from Northern Jutland found that nearly 50% of the women affected by malignant mesothelioma had been exposed to domestic asbestos through first-degree relatives [16]. In our study, one woman with documented familial exposure to asbestos was diagnosed with malignant mesothelioma versus none in the control group, which is too low to allow a statistical evaluation. However, this case, as part of the school cohort, also had environmental exposure.

Lung cancer was the second most common cancer among the female school cohort. The overall incidence of lung cancer was not elevated among the women from the school cohort. However, in our regression analysis, those subject to occupational or familial occupational asbestos exposure had a significantly increased hazard ratio. This is in line with a study from Casale Monferrato, where mortality from lung cancer was not increased in the population that was not exposed to occupational asbestos, but a large excess was found among women exposed to asbestos in connection with cement production; the attributable risk for female asbestos cement workers (and wives of asbestos workers) was 51.3% (95% CI 14.9–87.8) [30].

The SIRs for myeloma were significant; however, these results are difficult to interpret in view of the small number of cases and could be random findings.

Smoking is the major cause of several cancers [31]. Especially for lung cancer, for which the risk is increased [32]. We used a COPD diagnosis as a proxy for smoking because tobacco smoking is the most significant risk factor for the development of COPD [20]. Since not all who have smoked are diagnosed with COPD and not everyone diagnosed with COPD has a history of smoking [33], this approach is unlikely to control for all confounding factors related to individual smoking status, and some misclassification is unavoidable. In the regression analysis for female cancers and breast cancer, smoking did not increase the hazard ratio for any of the cancers.

This study has some unique features. First, we were able to identify a large cohort of female high-school students where all were tentatively asbestos-exposed simply by attending school and living nearby the asbestos cement factory. Secondly, we could estimate their occupational history, as well as the occupational asbestos exposure history of all their family members by using nearly complete Danish registries. This estimation of occupational asbestos exposure by registry linkage may result in the non-differential misclassification of a few cases. However, a large benefit of using this register-based study design is that we avoid recall bias. Third, we have a long follow-up period of more than 40 years, which is highly important since there is a long latency period for the known asbestos-related cancers [34]. Fourth, we had a 1:9 matched control population from the whole of Denmark (n = 54,200), and the family members of this population with available occupational asbestos exposure data.

A limitation of the estimation of environmental asbestos exposure is that it is based solely on 7th grade school attendance rather than on personal, monitored quantification. Furthermore, we have no information on any environmental asbestos exposure before or after the 7th grade school attendance, neither did we have information on women’s childbirth experience nor their income, both confounders to some of the female cancers.

Another limitation concerns the exposure to other carcinogens. Only asbestos exposure and tobacco smoking were taken into account in this study, and the role of co-exposure to other carcinogens cannot, therefore, be ruled out.

## 5. Conclusions

In conclusion, our study revealed an increased risk of cancer of the corpus uteri and malignant mesothelioma in the cohort of female former school children exposed to environmental asbestos. The increased risk may, for some part, have been due to lifestyle and behavioral characteristics. However, the potential effect of environmental asbestos exposure on some cancers cannot be discarded. For countries still using asbestos, we strongly urge companies, together with policymakers and health professionals, to recognize the risk of asbestos exposure to health, as well as low-level environmental asbestos exposure.

## Figures and Tables

**Figure 1 ijerph-19-02086-f001:**
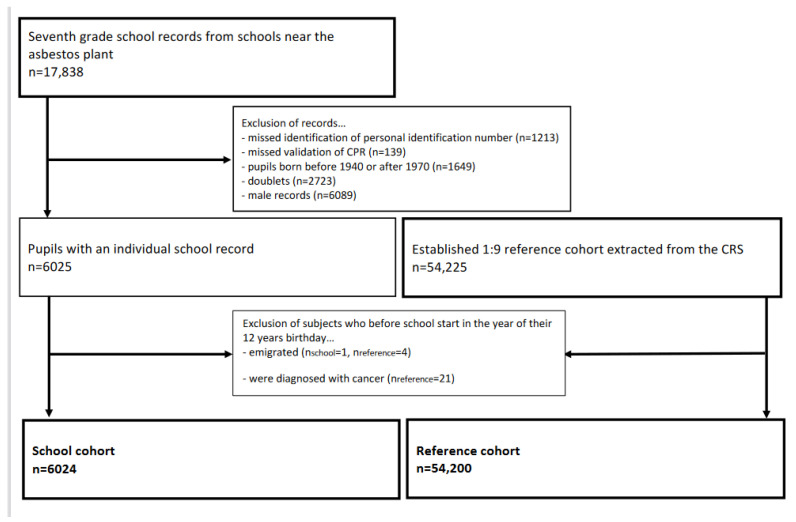
Flowchart of the establishment of the female cohorts.

**Table 1 ijerph-19-02086-t001:** Characteristics of the females from the Aalborg School Children Cohort and the nationwide reference cohort.

	School Cohort		Reference Cohort		
Characteristics	*n*		*n*		*p*-Value
Cohort size	6024		54,200		
Birth year (%)					
1940–1944	1190	(19.8)	10,707	(19.8)	
1945–1949	1452	(24.1)	13,067	(24.1)	
1950–1954	1335	(22.2)	12,008	(22.2)	
1955–1959	1071	(17.8)	9645	(17.8)	
1960–1964	754	(12.5)	6778	(12.5)	
1965–1970	222	(3.7)	1.995	(3.7)	
Person years of follow-up	297,637		2,583,877		
Median attained age (range)	63.0	(14.8–76.0)	62.2	(12.0–76.0)	0.000
Type of asbestos exposure (%)					0.000
Environmental exposure (school cohort) and unknown asbestos exposure (reference cohort)	4345	(72.1)	43,029	(79.4)	
Occupational asbestos exposure	138	(2.3)	1206	(2.2)	
Familial occupational asbestos exposure	1415	(23.5)	7753	(14.3)	
Occupational and familial occupational asbestos exposure	80	(1.3)	414	(0.8)	
No employment history records	46	(0.8)	1798	(3.3)	
Proxy for smoking (%)	355	(5.9)	2249	(4.2)	0.000

**Table 2 ijerph-19-02086-t002:** Characteristics of the female cancer cases from the Aalborg School Children Cohort and the nationwide reference cohort.

	School Cohort		Reference Cohort		
Characteristics	*n*		*n*	(%/Range)	*p*-Value
Cancer cases * (% of all cohort)	912	(15.1)	7991	(14.7)	
Birth-year (%)					0.523
1940–1944	295	(32.4)	2530	(31.7)	
1945–1949	262	(28.7)	2493	(31.2)	
1950–1954	187	(20.5)	1502	(18.8)	
1955–1959	106	(11.6)	923	(11.6)	
1960–1964	50	(5.5)	466	(5.8)	
1965–1970	12	(1.3)	77	(1.0)	
Median attained age (range)	63.9	(15.0–76.0)	64.4	(16.1–76.0)	0.122
Type of asbestos exposure (%)					0.000
Environmental exposure (school cohort) and unknown asbestos exposure (reference cohort)	672	(73.7)	6380	(79.8)	
Occupational asbestos exposure	23	(2.5)	209	(2.6)	
Familial occupational asbestos exposure	198	(21.7)	1277	(16.0)	
Occupational and familial occupational asbestos exposure	17	(1.9)	88	(1.1)	
No Supplementary Pension Fund Register data	2	(0.2)	37	(0.5)	
Proxy for smoking (%)	63	(6.9)	572	(7.2)	

* Individuals with at least one cancer.

**Table 3 ijerph-19-02086-t003:** Standardized incidence ratios for selected cancers among 6024 female former school children compared with a reference group of 54,200 females.

Cancer Site	O/E	SIR	95% CI
External female genital organs and vagina	11/6.48	1.70	0.94 to 3.06
Other and unspecified female genital organs	1/0.23	4.39	0.62 to 31.14
Breast	343/348.30	0.98	0.89 to 1.09
Ovary, fallopian tube and broad ligament	34/46.94	0.72	0.52 to 1.01
Cervix uteri	50/45.23	1.11	0.84 to 1.46
Corpus uteri	61/47.18	**1.29**	**1.01 to 1.66**
Mesothelioma	6/0.83	**7.26**	**3.26 to 16.15**
Lung, bronchus and trachea	121/109.28	1.11	0.93 to 1.32

Abbreviations: CI, confidence interval; E, expected number of cases; O, observed number of cases; SIR, standardized incidence ratio. Bold denotes statistically significant results.

**Table 4 ijerph-19-02086-t004:** Adjusted hazard ratios for the selected cancers in a cohort of 6024 female former school children and 54,200 female reference subjects.

Cancer Site			External Female Genital Organs and Vagina (*n* = 66)		
		*n*	Hazard Ratio	(95% CI)	*p* Value
Environmental asbestos exposure		11	1.74	(0.91–3.34)	0.091
Familial occupational asbestos exposure		12	-	-	-
Occupational asbestos exposure		3	-	-	-
Proxy for smoking		6	-	-	-
Cancer site			Breast (*n* = 3353)		
		*n*	Hazard ratio	(95% CI)	*p* Value
Environmental asbestos exposure		343	1.00	(0.89–1.12)	0.987
Familial occupational asbestos exposure		574	1.07	(0.98–1.17)	0.145
Occupational asbestos exposure		127	1.03	(0.86–1.23)	0.747
Proxy for smoking		191	0.91	(0.78–1.05)	0.184
Cancer site			Ovary, fallopian tube and broad ligament (*n* = 438)		
		*n*	Hazard ratio	(95% CI)	*p* Value
Environmental asbestos exposure		34	0.76	(0.54–1.08)	0.130
Familial occupational asbestos exposure		60	1.31	(0.99–1.72)	0.055
Occupational asbestos exposure		15	0.25	(0.15–0.41)	0.000
Proxy for smoking		22	0.89	(0.58–1.37)	0.593
Cancer site			Cervix uteri (*n* = 438)		
		*n*	Hazard ratio	(95% CI)	*p* Value
Environmental asbestos exposure		50	1.11	(0.83–1.50)	0.472
Familial occupational asbestos exposure		78	1.08	(0.85–1.39)	0.525
Occupational asbestos exposure		24	1.55	(1.03–2.35)	0.036
Proxy for smoking		20	0.75	(0.48–1.18)	0.215
Cancer site			Corpus uteri (*n* = 474)		
		*n*	Hazard ratio	(95% CI)	*p* Value
Environmental asbestos exposure		61	1.32	(1.02–1.75)	0.045
Familial occupational asbestos exposure		77	1.00	(0.78–1.28)	0.988
Occupational asbestos exposure		16	0.91	(0.55–1.50)	0.713
Proxy for smoking		/	/	/	/
Cancer site			Mesothelioma (*n* = 13)		
		*n*	Hazard ratio	(95% CI)	*p* Value
Environmental asbestos exposure		6	7.41	(2.49–22.06)	0.000
Familial occupational asbestos exposure		1	-	-	-
Occupational asbestos exposure		0	-	-	-
Proxy for smoking		/	/	/	/
Cancer site			Lung, bronchus, and trachea (*n* = 1064)		
		*n*	Hazard ratio	(95% CI)	*p* Value
Environmental asbestos exposure	121		1.04	(0.86–1.26)	0.677
Familial occupational asbestos exposure	216		1.23	(1.06–1.43)	0.007
Occupational asbestos exposure	57		1.38	(1.06–1.80)	0.018
Proxy for smoking	200		3.55	(3.05–4.15)	0.000

- Indicates no/too few observations for adjustment analysis. / Indicates a risk factor not relevant for particular cancer site. Bold denotes statistically significant results.

## Data Availability

Not applicable.

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
