# Peer review of "A Cohort Study on Cancer Incidence among Women Exposed to Environmental Asbestos in Childhood with a Focus on Female Cancers, including Breast Cancer"

_ijerph, 2022, doi:10.3390/ijerph19042086_

Round 1

Reviewer 1 Report

The authors wanted to examine the risk of cancer in school girls who were exposed to environmental asbestos in Denmark.  These girls were matched to a reference cohort.  The authors claim the girls exposed to environmental asbestos were at an increased risk of mesothelioma and cervical cancer.

Major Points:

The authors show in Table S.1 the Cancer Incidence among the 6024 school girls when compared to the reference group.  I think this table is more effective as part as the overall data as Table 5.

Lines 188-191.  Please provide a little more prose in the results so as to guide the reader about what you are trying to show in Table 3.

Table 4 needs a proper title.  It appears that the authors might have merged the prose and the title.

In the discussion, the authors do speak to the limitations of the study.  One area I thought might be worth mentioning, as it pertains to cervical cancer, is the prevalence of HPV in the population.  Can the authors say that the increase in cervical cancer is due to asbestos, or are other risk factors at play.

I do not agree with the analysis on smoking as documented by COPD only.  I do not feel that this is an adequate correlation of smoking, as we all know many people who smoked or have smoked an not been diagnosed with COPD.  I would either reword the section 2.4 and paragraph lines 328-335 or omit.

Minor points:

The formatting of the references is inconsistent.  Some are in brackets while in other cases they are superscript.

Omit table references in the methodology section (lines 163) and (167).

Omit " " in "other urinary organ" (lines 215 and 325)

Author Response

Dear Reviewer. 

Thank you for your comments and recommendations.

Please find the your comments and our reply in the attached file. 

Kind regards

Sofie Bünemann Dalsgaard

Reviewer 2 Report

The aim is stated clear. Dalsgaard et al. stated clearly what study found and how they did it. The title is informative and relevant. The research question also justified given what is already known about the topic.

The variables are well defined and measured appropriately. The study methods are valid and reliable. The data is presented in an appropriate way. The text in the results add to the data and it is not repetitive. Statistically significant results are clear. Results are discussed from different angles and placed into context without being overinterpreted.

The conclusions answer the aim of the study. The conclusions are supported by references and own results. The limitations of the study are not fatal, but they are opportunities to inform future research.

Specific comments on weaknesses of the article and what could be improved:

Major points  - none

Minor points

  1. Could you state your recommendations based on your study - how "policymakers and health professionals to recognize the risk to
    health from asbestos exposure – also low level environmental asbestos exposure"

Author Response

(The authors gave the same response as above.)

Reviewer 3 Report

This is a cohort study investigating the incidence of female cancer in early childhood asbestos exposure. The manuscript was well prepared. Moreover, the cited and their discussion are presented in good style. However, authors have a point that needs to be addressed.

  1. Is the childbirth experience of the subject taken into consideration?
  2. Did you consider the confounding of the subject's income.

    If you have not considered the above items, please add a discussion or clearly state it as a limitation.

Author Response

(The authors gave the same response as above.)
